# Survival Comparison of Different Operation Types for Middle Bile Duct Cancer: Bile Duct Resection versus Pancreaticoduodenectomy Considering Complications and Adjuvant Treatment Effects

**DOI:** 10.3390/cancers16020297

**Published:** 2024-01-10

**Authors:** Soo Yeun Lim, Hani Jassim Alramadhan, HyeJeong Jeong, Hochang Chae, Hyeong Seok Kim, So Jeong Yoon, Sang Hyun Shin, In Woong Han, Jin Seok Heo, Hongbeom Kim

**Affiliations:** 1Division of Hepatobiliary-Pancreatic Surgery, Department of Surgery, Samsung Medical Center, Sungkyunkwan University School of Medicine, Seoul 06351, Republic of Korea; sooyeun.lim.x@gmail.com (S.Y.L.); hjeong.sophia@gmail.com (H.J.); hochang.chae@samsung.com (H.C.); hs0853.kim@samsung.com (H.S.K.); sojeong.yoon@samsung.com (S.J.Y.); iw.han@samsung.com (I.W.H.); jinseok.heo@samsung.com (J.S.H.); 2Department of Surgery, King Fahad Hospital, Hofuf 36441, Saudi Arabia; dr.hani-jar@hotmail.com

**Keywords:** cholangiocarcinoma, middle bile duct cancer, margin status, pancreaticoduodenectomy, bile duct resection, adjuvant treatment

## Abstract

**Simple Summary:**

This study aimed to investigate the appropriate treatment according to the operation type, margin status, complications, and adjuvant treatment for middle bile duct cancer. The age, preoperative CA19-9 level, T stage, and N stage affected the overall survival; however, the operation type, margin status, complication, or adjuvant treatment did not. There were no significant differences in the adjuvant treatment ratio according to complications and operation type (bile duct resection vs. pancreaticoduodenectomy). Patients who underwent PD with R0 resection and could not undergo chemotherapy because of complications reported better survival rates than those who underwent BDR with R1 resection after adjuvant treatment. Therefore, the surgeon should secure an R0 margin to achieve the best survival outcome.

**Abstract:**

Background: Margin status is one of the most significant prognostic factors after curative surgery for middle bile duct (MBD) cancer. Bile duct resection (BDR) is commonly converted to pancreaticoduodenectomy (PD) to achieve R0 resection. Additionally, adjuvant treatment is actively performed after surgery to improve survival. However, the wider the range of surgery, the higher the chance of complications; this, in turn, makes adjuvant treatment impossible. Nevertheless, no definitive surgical strategy considers the possible complication rates and subsequent adjuvant treatment. We aimed to investigate the appropriate surgical type considering the margin status, complications, and adjuvant treatment in MBD cancer. Materials and Methods: From 2008 to 2017, 520 patients diagnosed with MBD cancer at the Samsung Medical Center were analyzed retrospectively according to the operation type, margin status, complications, and adjuvant treatment. The R1 group was defined as having a carcinoma margin. Results: The 5-year survival rate for patients who underwent R0 and R1 resection was 54.4% and 33.3%, respectively (*p =* 0.131). Prognostic factors affecting the overall survival were the age, preoperative CA19-9 level, T stage, and N stage, but not the operation type, margin status, complications, or adjuvant treatment. The complication rates were 11.5% and 29.8% in the BDR and PD groups, respectively (*p* < 0.001). We observed no significant difference in the adjuvant treatment ratio according to complications (*p* = 0.675). Patients with PD who underwent R0 resection and could not undergo chemotherapy because of complications reported better survival rates than those with BDR who underwent R1 resection after adjuvant treatment (*p* = 0.003). Conclusion: The survival outcome of patients with R1 margins who underwent BDR did not match those with R0 margins after PD, even after adjuvant treatment. Due to improvements in surgical techniques and the ability to resolve complications, surgical complications exert a marginal effect on survival. Therefore, surgeons should secure R0 margins to achieve the best survival outcomes.

## 1. Introduction

Cholangiocarcinoma (CCC) is a rare aggressive malignancy. The incidence of intrahepatic cholangiocarcinoma ranges from 0.44 to 1.18 cases per 100,000 individuals, while the incidence of extrahepatic cholangiocarcinoma ranges from 0.95 to 1.02 cases per 100,000 individuals during the four-decade period [1]. It can be classified into intrahepatic CCC and extrahepatic CCC according to its anatomic location within the biliary tree [2]. Extrahepatic CCC can be further subdivided into hilar CCC (50%), middle CCC (19%), and distal (25%) CCC [3,4,5,6]. Surgical resection with negative margins provides patients with the highest chance of cure [5,7]. The prognosis is highly dependent on the resection margin status. Patients with a positive resection margin reported dismal survival outcomes [8].

The surgical approach for extrahepatic CCC is based on the anatomical location and extent of the tumor. Bile duct resection (BDR) with concomitant liver resection is adopted for hilar CCC types III to IV, whereas pancreaticoduodenectomy (PD) is performed for distal CCC. However, the surgical management of middle bile duct (MBD) cancer, which extends from the hilar confluence to the distal lesion and does not reach the upper border of the pancreas, is still debatable [9]. Isolated MBD cancer without infiltration into the proximal or distal bile duct is rare. This is because bile duct cancers tend to spread longitudinally along the bile duct wall [9,10]. Several studies have advocated liver resection in hilar CCC types I and II to increase the rate of R0 margins and improve survival outcomes [11]. However, other studies have advocated PD in middle and distal CCC for a similar reason. Limited BDR is an acceptable approach for MBD and may result in similar oncological outcomes upon achieving negative margins [9,12,13,14]. However, these studies were limited by small sample sizes because of their rare prevalence. Extended BDR with major hepatectomy or PD is associated with a higher risk of morbidity and mortality [15]. PD is associated with several complications, such as postoperative pancreatic fistula, post-pancreatectomy hemorrhage, and surgical site infection, which may delay or prevent adjuvant treatment [16,17]. In the era of modern chemotherapy, several authors have reported on the beneficial effects of adjuvant treatment in extrahepatic CCC [18,19,20]. Some studies have reported similar survival outcomes between patients with an R0 margin without chemotherapy and those with an R1 margin who received chemotherapy [19,21]. Thus, modern adjuvant treatments could reverse the negative effects of the R1 margin in extrahepatic CCC; therefore, extended resection, which includes PD to achieve the R0 margin, may be unnecessary because the higher risk of postoperative complications associated with PD may delay adjuvant treatment.

There is no definite surgical strategy considering the possible complication rate and subsequent adjuvant treatment. We aimed to investigate the appropriate operation type, particularly between PD and BDR, considering the margin status, complication, and adjuvant treatment in MBD cancer.

## 2. Material and Methods

### 2.1. Patients and Surgical Procedures

Figure 1 depicts the patient flowchart. From 2008 to 2017, 581 patients who were diagnosed with MBD cancer at the Samsung Medical Center were analyzed retrospectively according to the operation type, margin status, complications, and adjuvant treatment. Nine patients who underwent non-curative surgery, including surgery termination because of occult metastasis or palliative resection (R2 resection margin), were excluded. Moreover, patients with BDR with proximal margin R1 (*n* = 8) or PD with proximal margin R1 (*n* = 8) were excluded. This study was approved by the Institutional Review Board of the Samsung Medical Center (IRB No. 2022-11-098). Also the study was analyzed retrospectively by medical records, there is no copyright issue. A total of 520 patients were enrolled in this study. The patients were divided into two groups based on the type of operation as follows: 131 in the BDR group and 389 in the PD group.

### 2.2. Diagnosis and Definition of Surgical Margins

We evaluated the location and extent of the tumor along the biliary tract using imaging studies, including an enhanced computed tomography scan, ultrasonography, magnetic resonance imaging, and cholangiography.

The surgical procedures were decided by each attending surgeon based on the tumor location and extension, margin status of the intraoperative frozen section, and perioperative risk. After excluding distant metastasis by exploration-lapa, a regional lymphadenectomy was performed at the right side of the celiac artery. All tissues in the hepatoduodenal ligament, except those in the portal vein and hepatic artery, were removed using skeletonization of the hepatoduodenal ligament. BDR was performed with bismuth type I and II hilar CCC and suprapancreatic distal CCC for tumors located in the distal bile duct. Intra-operative bile duct frozen sections of the proximal (hepatic) and/or distal (duodenal) ductal margins were obtained from all patients. For a positive distal ductal margin, the patients underwent an additional resection of the intrapancreatic bile duct or PD. For a positive proximal ductal margin, the patients underwent an additional resection of the hepatic duct or hepatectomy.

The specimens were submitted to the Department of Pathology at the Samsung Medical Center. Cross-sections of the proximal and distal bile duct margins were examined by an experienced hepatobiliary pathologist without prior knowledge of any previous diagnosis or clinical details. The margins were classified into two categories, namely, negative margin R0 and microscopic positive margin R1. Surgical margins other than the ductal margins were labeled as radial margins. We used the TNM staging system (seventh edition) of the American Joint Committee on Cancer to describe the histological findings.

### 2.3. Comparison of the Clinicopathological Variables and Follow-Up

We evaluated the clinicopathological variables, including age, sex, preoperative carbohydrate antigen 19-9 (CA 19-9), postoperative hospital stay, postoperative complications, location of a positive margin, histological grade, and adjuvant treatment. Major complications were defined as Clavien–Dindo grade >3 within 90 days after surgery.

Postoperative follow-up was performed regularly for all patients every 3 to 6 months. Cross-sectional imaging was used to detect disease recurrence, which was classified as local (resection margin, biocentric anastomosis, regional lymph nodes, and porta-hepatis) or systemic (intrahepatic, peritoneal, or extra-abdominal sites). The follow-up period was defined as the interval between the date of surgery and the last follow-up. Overall survival (OS) was defined as the period from the date of surgery to the date of death. Medical records were reviewed retrospectively to determine the causes of death.

### 2.4. Adjuvant Treatment

Patients with R1 margins and stage 2 cancer were referred to a medical oncologist for adjuvant treatment. A medical oncologist makes the decision to administer adjuvant treatment, the type of treatment, and the regimen based on the patient’s condition.

### 2.5. Statistical Analysis

Continuous variables are expressed as means and standard deviations. Categorical variables are expressed as numbers and proportions. Median values and interquartile ranges (IQRs) are used to describe the tumor markers. Student’s *t*-test and x^2^ test were used to compare the categorical variables. We performed an independent t-test or Mann–Whitney U test to compare the continuous variables. The OS and disease-free survival (DFS) were analyzed using the Kaplan–Meier method. We used the log-rank test to analyze differences in the survival curves. Factors independently affecting survival were identified using a Cox proportional hazards model. *p*-values < 0.05 were considered statistically significant. A multivariate analysis of independent prognostic factors for OS was identified using the Cox proportional hazard model. All tests were two-sided, and a p-value under 0.1 was considered statistically significant. There were no missing data in this study. IBM SPSS Statistics for Windows version 27 (IBM Corp., Armonk, NY, USA) was used for data analysis.

## 3. Results

### 3.1. Demographics and Margin Status

A total of 520 patients were included in this study. Table 1 summarizes the patient demographics and clinicopathological data. A total of 342 patients were men (65.8%) and 178 were women (34.2%), with a median age of 66.7 (±8.4) years. We classified 131 patients in the BDR group, of which 6 demonstrated distal margin positivity (4.6%). The PD group comprised 389 patients. The patients who underwent PD were younger than those who underwent BDR (67.9 years vs. 66.3 years, *p* = 0.044). The PD group reported a more advanced T-stage stage than the BDR group (T3/4: 6.9% vs. 16.2%, *p* = 0.003). The average numbers of retrieved lymph nodes were 13.5 and 19.3 in the BDR and PD groups, respectively (*p* < 0.001). However, we observed no significant difference in the N stage between the groups (*p* = 0.119). The postoperative hospitalization was significantly longer in the PD group (9.0 vs. 12.0, *p* < 0.001). Four patients (66.7%) in the R1 margin group and 113 (22.0%) in the R0 margin group received adjuvant treatment (*p* = 0.025). Appendix A summarizes the clinical course of margin positive. Despite the positive margin, two patients did not undergo an extended operation due to the patient’s comorbidities.

### 3.2. Survival Analysis

We observed no significant difference in the OS or DFS between the groups (estimated 5-year OS: 54.7% vs. 54.0%, *p* = 0.594, estimated 5-year DFS: 34.4% vs. 39.0%, *p* = 0.803) (Figure 2). The survival graph demonstrated that the R0 group was superior to the R1 group; nonetheless, there were no significant differences between the groups regarding the survival (estimated 5-year OS: 54.4% vs. 33.3%, *p* = 0.131; estimated 5-year DFS: 38.4% vs. 0%, *p* = 0.127) (Figure 3).

### 3.3. Prognostic Factors for Survival in Mid-Bile Duct Cancer

The multivariate Cox proportional hazards analysis suggested that the patients age (Hazard ratio (HR) 2.187, 95% CI 1.689–2.832, *p* < 0.001), preoperative CA19-9 (HR 1.309, HR 1.026–1.672, *p* = 0.030), T-stage (HR 1.677, 95% CI 1.234–2.279, *p* = 0.001), and N-stage (HR 2.013, 95% CI 1.554–2.609, *p* < 0.001) were the independent prognostic factors for overall survival (Table 2). The surgical type, complications, adjuvant treatment, and resection margins were not included as the risk factors. In the DFS analysis, the age, preoperative CA19-9 level, T stage, and N stage were the independent risk factors (Appendix A).

### 3.4. Postoperative Complications

Of the 520 patients, 131 (25.2%) presented with major postoperative complications. The major complication rate was significantly lower in the BDR group (11.5%) than in the PD group (29.8%) (*p* < 0.001). The common complications comprised surgical site infection (6.1%), chylous ascites (4.6%), intra-abdominal abscess (3.1%) in the BDR group, and POPF (19.3%), chylous ascites (8.5%), and SSI (7.2%) in the PD group.

### 3.5. Adjuvant Treatment Rate According to Complications

Table 3 summarizes the adjuvant treatment rates according to the operation type and complications. We observed no significant difference in the rate of adjuvant chemotherapy between the BDR and PD groups (9.2% vs. 7.7%, *p* = 0.599). The rates of adjuvant treatment, including chemotherapy and radiotherapy, were comparable between the groups, without major complications. (*p* = 0.760). Patients with complications after both PD and BDR demonstrated no difference in the adjuvant treatment rate (*p* = 0.193).

### 3.6. Survival Outcomes According to the Margin, Operation Type, Complication, and Adjuvant Treatment

We compared the survival outcomes between the BDR and PD groups that were stratified based on the surgical margins, major complications, and adjuvant treatment (Figure 4). First, the PD group with R0 margins reported better survival than the BDR group with R1 margins that received adjuvant treatment (*p* = 0.001) (Figure 4A). Patients who underwent PD with R0 margins and reported major complications demonstrated significantly better survival outcomes than those who underwent BDR with R1 margins and received adjuvant therapy (*p* = 0.006) (Figure 4B). Patients who could not receive chemotherapy because of postoperative complications after PD with an R0 margin demonstrated better survival than those who underwent BDR with an R1 margin and received adjuvant therapy (*p* = 0.022) (Figure 4C).

## 4. Discussion

CCC is a rare tumor that accounts for only 3% of all gastrointestinal malignancies [21,22]. It has been classified into intrahepatic CCC and extrahepatic CCC, which includes both perihilar and distal CCC, according to its anatomical location. Each category has different oncological behaviors, prognoses, and surgical approaches for management [23]. Surgical resection provides patients with only a potential cure [22,24]. The majority of extrahepatic CCC are of the perihilar type, termed Klatskin tumors, accounting for 50% of the cases, followed by distal and peri-ampullary CCC, which account for 20% to 30% of the cases [2,23]. CCC limited to the middle third of the bile duct is rare because it tends to spread longitudinally along the bile duct wall [9,10].

Limited BDR is an oncologically acceptable approach that yields survival outcomes similar to those in PD; however, the number of cases is relatively small to provide strong evidence [9,12,14,25]. We analyzed a large number of MBD cancer cases and obtained similar results. We observed no statistically significant differences in the OS and DFS between the BDR and PD groups. The type of surgical procedure was not an independent factor influencing the survival or recurrence. In contrast, patients with BDR reported significantly shorter postoperative hospitalization and lower rates of postoperative complications. Thus, BDR can be a safe approach for patients with MBD cancer upon achieving an R0 margin at both proximal and distal margins.

The resection margin is one of the strongest prognostic factors for survival [26]. The rate of R0 resection margin after BDR is significantly lower than that after PD, ranging from 38% to 65% [14,25,27,28]. In our study, the rate of R0 margin after BDR was higher than that reported previously (89.9%). In a multicenter collaborative study between Japan and Korea, Hayashi et al. [28] demonstrated that the rate of R0 after BDR decreased with a higher T-stage. The R0 rate after BDR was significantly lower than that after PD at T2 (56.1% vs. 67.4%, *p* = 0.340) and T3 (35.3% vs. 66.3%, *p* = 0.072). Thus, patient selection, surgical technique, and the use of intraoperative frozen sections are important factors. In our study, six patients demonstrated distal positive margins after BDR; Appendix A describes their clinical course. Four patients underwent R1 resection because of altered results after intraoperative frozen-section biopsy. Two patients demonstrated a positive margin and did not undergo an extended operation because of their comorbidities and condition. Of the six patients, only two received adjuvant radiotherapy.

The number of LN retrieved after BDR is significantly lower than that after PD. However, the rate of LN metastasis was similar between the groups [25,28]. This is because the BDR procedure could not retrieve the peripancreatic lymph nodes (station 13,17) and superior mesenteric artery lymph nodes (station 14) [25]. Akita et al. [25] reported that one-third of the patients with MBD who underwent PD presented with positive lymph node metastasis around the pancreatic head. Thus, BDR should be contraindicated upon suspecting lymph node metastasis at stations 13, 14, and 17.

PD is a major surgical procedure associated with a high risk of perioperative morbidity in up to 58% of the cases [29,30]. Pugalenthi et al. [29] mentioned that the rate of major complications was 22% in 596 patients who underwent PD for pancreatic cancer; these complications did not affect the overall survival (*p* = 0.948). Wu et al. [31] investigated the effects of postoperative complications on adjuvant therapy after PD. The complications did not influence the adjuvant therapy; however, they delayed the initiation of adjuvant treatment. In our study, we observed no significant difference in the rate of adjuvant chemotherapy between the BDR and PD groups (*p* = 0.599). Major complications did not result in a significant difference in the adjuvant treatment between the groups (*p* = 0.193).

The role of adjuvant therapy in patients with resectable bile duct cancer remains poorly defined. Several studies have illustrated the beneficial effects of adjuvant chemotherapy in resectable bile duct cancer [18,19,20,32]. The British randomized control trial [32] demonstrated a longer median overall survival in the capecitabine group than that in the observation group after curative intended surgery for biliary tract cancer. Adjuvant chemotherapy in cases with an R1 positive margin results in similar survival outcomes than in those with R0 margins without chemotherapy [19,33]. Lee et al. [19] demonstrated that the survival rate after R1 resection and adjuvant gemcitabine-based chemotherapy was similar to that after R0 resection without adjuvant treatment. In addition, Lee et al. [33] reported comparable survival rates between patients with R1 resection with adjuvant chemoradiotherapy and those with R0 resection without adjuvant treatment. These studies had a relatively low number of cases, and the cohort was heterogeneous in that they included patients with gallbladder cancer and all types of bile duct cancer, who may demonstrate different biological behaviors. In this study, the R0 group without adjuvant therapy reported significantly better survival rates than the R1 group with adjuvant treatment. Adjuvant therapy and resection were not independent prognostic factors in the multivariate analysis; however, achieving R0 resection should be a priority in all cases.

This study has some limitations. It was conducted at a large tertiary center; nonetheless, fewer patients were stratified according to the multimodal factors, including the operation, complications, and adjuvant treatment. In addition, the study was based on a retrospective analysis, which could not provide strong evidence. Finally, our conclusion emphasized the surgeon’s ability to secure a negative margin status. However, because of the small number of patients who underwent R1 resection, the margin status was not included as an independent risk factor, thus necessitating a sequential multicenter study.

Nevertheless, in the absence of a precise surgical strategy for middle bile duct cancer and the effect of adjuvant treatment, our study is valuable because we compared BDR and PD, considering their complications and subsequent treatment effects. To the best of our knowledge, this is the first study to comprehensively compare the survival with adjuvant treatment, complication, and margin status.

## 5. Conclusions

With improvements in surgical techniques and the modalities available to manage postoperative complications, complications exert marginal effects on the survival outcome, and they do not delay nor prevent adjuvant treatment. BDR can be an acceptable approach for selecting patients upon achieving an R0 margin with additional adjuvant treatment. Therefore, surgeons should secure the R0 margins as much as possible.

## Figures and Tables

**Figure 1 cancers-16-00297-f001:**
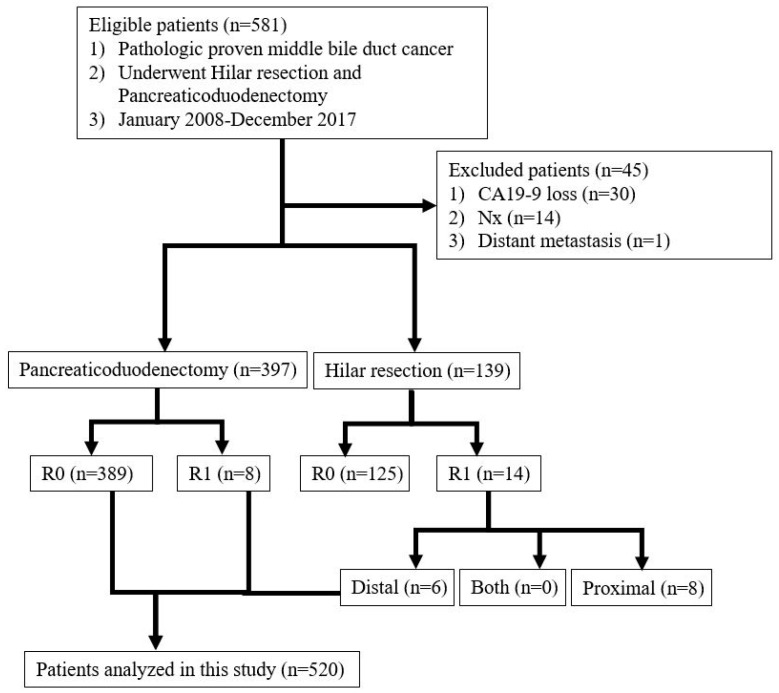
Patients selection.

**Figure 2 cancers-16-00297-f002:**
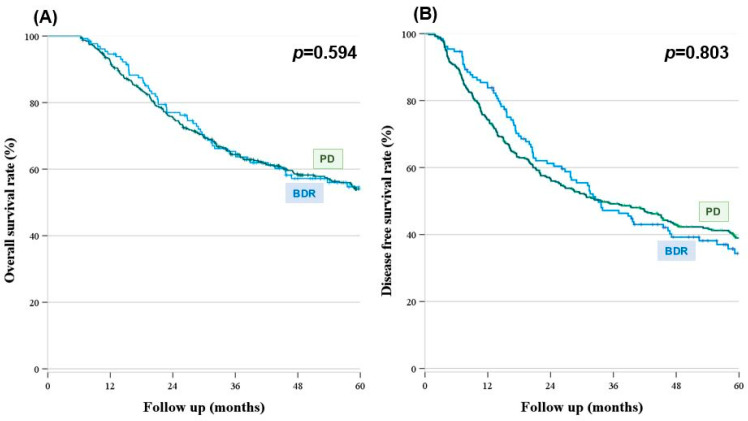
Survival outcomes according to operation type (**A**) Overall survival (**B**) Disease-free survival (*n* = 520). (**A**) The median overall survival was not reached in either the BDR or PD groups. The estimated 5-year survival rates were 54.7% and 54.0% in the BDR and PD groups, respectively, with no statistically significant differences between the two groups (*p* = 0.594). (**B**) Graph of disease-free survival by operation type demonstrates that there were no differences between the BDR and PD groups (Median BDR, 33.6% vs. PD, 33.8%, *p* = 0.803).

**Figure 3 cancers-16-00297-f003:**
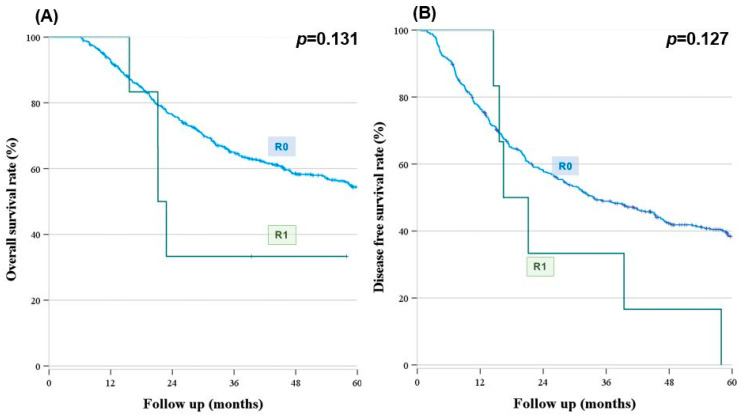
Survival outcomes according to margin status. (**A**) Overall survival (**B**) Disease-free survival (*n* = 520). (**A**) The overall survival graph by margin status demonstrated that there were no statistical differences between the R0 and R1 groups (estimated 5-year OS; R0 54.4% vs. R1 33.3%, *p* = 0.131). (**B**) Median disease-free survival was 33.6 months in the R0 group and 16.4 months in the R1 group. However, there were no statistical differences between the two groups (*p* = 0.127).

**Figure 4 cancers-16-00297-f004:**
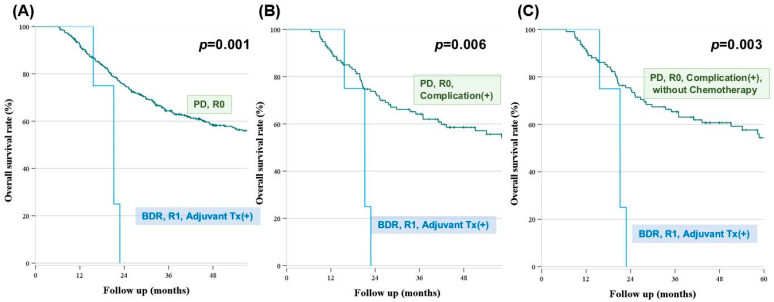
Survival outcomes according to the margin, OP type, presence of complication, and Adjuvant Tx. (**A**) The median survival of patients with an R1 margin who underwent BDR and received adjuvant treatment was 21.2 months. For patients who underwent PD with an R0 margin, the estimated 5-year survival was 54.0%, and the median survival time was not reached. (**B**) Among patients who underwent PD with an R0 margin and experienced complications, the estimated 5-year survival rate was 52.5%, and the median survival was not reached. (**C**) For patients who underwent PD with an R0 margin but were unable to receive chemotherapy due to complications, the estimated 5-year survival rate was 51.3%, and the median overall survival was not reached.

**Table 1 cancers-16-00297-t001:** Demographics and Clinical Characteristics.

Variables	*n* (%) or Mean (SD)	Total (*n* = 520)	BDR (*n* = 131)	PD (*n* = 389)	*p*-Value	R0 (*n* = 514)	R1 (*n* = 6)	*p*-Value
Sex (M:F)		342:178	91:40	251:138	0.303	338:176	4:2	0.963
Age		66.7 (8.4)	67.9 (8.1)	66.3 (8.5)	0.044	66.6 (8.4)	72.8 (2.7)	0.002
BMI		23.4 (3.1)	23.6 (3.4)	23.4 (3.0)	0.491	23.4 (3.1)	22.0 (1.3)	0.043
ASA	I	82 (15.8)	17 (13.0)	65 (16.7)	0.359	81 (15.8)	1 (16.7)	0.570
	II	395 (76.0)	100 (76.3)	295 (75.8)		391 (76.1)	4 (66.7)	
	III/IV	43 (8.3)	14 (10.7)	29 (7.5)		42 (8.2)	1 (16.7)	
CA 19-9 median (IQR)		34.0 (16.4–110.9)	29.2 (12.5–74.0)	36.6 (17.2–123.3)	0.199	34.0 (16.4–110.7)	32.5 (7.4–190.4)	0.005
T-stage	T1	201 (38.7)	45 (34.4)	156 (40.1)	0.003	200 (38.9)	1 (16.7)	0.247
	T2a/T2b	247 (47.5)	77 (58.8)	170 (43.7)		244 (47.5)	3 (50.0)	
	T3/T4	72 (13.8)	9 (6.9)	63 (16.2)		70 (13.6)	2 (33.3)	
N-stage	N0	356 (68.5)	95 (72.5)	261 (67.1)	0.119	353 (68.7)	3 (50.0)	0.547
	N1	134 (25.8)	33 (25.2)	101 (26.0)		131 (25.5)	3 (50.0)	
	N2	30 (5.8)	3 (2.3)	27 (6.9)		30 (5.8)	0 (0)	
Retrieved Lymph node		17.8 (8.6)	13.5 (6.6)	19.3 (8.7)	<0.001	17.9 (8.5)	10.3 (8.8)	0.090
Postop hospital days median (IQR)		11.0 (9.0–15.0)	9.0 (8.0–11.0)	12.0 (10.0–17.0)	<0.001	11.0 (9.0–15.0)	10.5 (8.75–16.5)	0.477
Major complication	No	389 (74.8)	116 (88.5)	273 (70.2)	<0.001	384 (74.7)	5 (83.3)	0.628
	Yes	131 (25.2)	15 (11.5)	116 (29.8)		130 (25.3)	1 (16.7)	
Adjuvant treatment	No	403 (77.5)	98 (74.8)	305 (78.4)	0.394	401 (78.0)	2 (33.3)	0.025
	Yes	117 (22.5)	33 (25.2)	84 (21.6)		113 (22.0)	4 (66.7)	
Adjuvant Chemotherapy	No	478 (91.9)	119 (90.8)	359 (92.3)	0.599	474 (92.2)	4 (66.7)	0.078
	Yes	42 (8.1)	12 (9.2)	30 (7.7)		40 (7.8)	2 (33.3)	
Adjuvant Radiotherapy	No	417 (80.2)	105(80.2)	312 (80.2)	0.989	413 (80.4)	4 (66.7)	0.340
	Yes	103 (19.8)	26 (19.8)	77 (19.8)		101 (19.6)	2 (33.3)	

BDR, Bile duct resection; PD/PPPD, pancreaticoduodenectomy; SD, Standard deviation.

**Table 2 cancers-16-00297-t002:** Prognostic Factors for Overall Survival in Patients with Middle Bile Duct Cancer.

Variable		Patients (*n*)	5Y OS (%)	Univariate Analysis			Multivariate Analysis		
				HR	95% CI	*p*-Value	HR	95% CI	*p*-Value
Sex	male/female	342/178	52.5/57.2	0.871	0.676–1.122	0.285			
Age	≤65/>65	229/291	65.4/44.7	2.087	1.619–2.690	<0.001	2.187	1.689–2.832	<0.001
BMI	≤25/>25	378/142	50.9/63.2	0.769	0.579–1.021	0.070	0.837	0.629–1.113	0.221
ASA score	I	82	57.1			0.176			
	II	395	54.6	1.133	0.817–1.572	0.455			
	III/IV	43	43.7	1.567	0.969–2.534	0.067			
Preop CA19-9	≤35/>35	265/255	62.3/45.9	1.539	1.210–1.958	<0.001	1.309	1.026–1.672	0.030
T-stage	T1 & 2/T3 & 4	448/72	58.4/28.0	2.361	1.761–3.164	<0.001	1.677	1.234–2.279	0.001
N-stage	N(−)/N(+)	359/161	62.4/35.5	2.154	1.689–2.747	<0.001	2.013	1.554–2.609	<0.001
Operation	BDR/PD	131/389	54.7/54.0	0.929	0.707–1.219	0.594			
Resection margin	R0/R1	514/6	54.4/33.3	2.108	0.783–5.678	0.131			
Complications ^a^	no/yes	389/131	55.3/50.3	1.046	0.791–1.383	0.752			
Adjuvant treatment	no/yes	403/117	55.6/49.1	1.077	0.806–1.437	0.617			

^a^ Major complication indicated Clavien-Dindo grade ≥ 3. OS, overall survival; PD/PPPD, pancreaticoduodenectomy/pylorus-preserving pancreaticoduodenectomy; BDR, Bile duct resection.

**Table 3 cancers-16-00297-t003:** Adjuvant Treatment Rates According to the Presence of Complications.

*n* (%)	Treatment	Total (*n* = 520)	BDR (*n* = 131)	PD (*n* = 389)	*p*-Value
Total patients	Adjuvant therapy	117 (22.5)	33 (25.2)	84 (21.6)	0.394
	Chemotherapy	42 (8.1)	12 (9.2)	30 (7.7)	0.599
	Radiotherapy	103 (19.8)	26 (19.8)	77 (19.8)	0.989
	Chemo & Radiotherapy	28 (5.4)	5 (3.8)	23 (5.9)	0.502
Major complication (+)		Total (*n* = 131)	BDR (*n* = 15)	PD (*n* = 116)	*p*-value
	Adjuvant therapy	27 (20.6)	5 (33.3)	22 (19.0)	0.193
	Chemotherapy	6 (4.6)	1 (6.7)	5 (4.3)	0.525
	Radiotherapy	26 (19.8)	5 (33.3)	21 (18.1)	0.177
	Chemo & Radiotherapy	5 (3.8)	1 (6.7)	4 (3.4)	0.461
Major complication (−)		Total (*n* = 389)	BDR (*n* = 116)	PD (*n* = 273)	*p*-value
	Adjuvant therapy	90 (23.1)	28 (24.1)	62 (22.7)	0.760
	Chemotherapy	36 (9.3)	11 (9.5)	25 (9.2)	0.919
	Radiotherapy	77 (19.8)	21 (18.1)	56 (20.5)	0.585
	Chemo & Radiotherapy	23 (5.9)	4 (3.4)	19 (7.0)	0.241

Major complication indicated Clavien-Dindo grade ≥ 3. PD/PPPD, pancreaticoduodenectomy/pylorus-preserving pancreaticoduodenectomy; BDR, Bile duct resection.

## Data Availability

The data set is not publicly available due to privacy.

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
