# Peer review of "Survival Comparison of Different Operation Types for Middle Bile Duct Cancer: Bile Duct Resection versus Pancreaticoduodenectomy Considering Complications and Adjuvant Treatment Effects"

_cancers, 2024, doi:10.3390/cancers16020297_

Round 1
Reviewer 1 Report
Comments and Suggestions for Authors
The authors present their experience with the different types of surgery for bile duct cancer, considering the margin status, complications, and adjuvant therapy.
The manuscript is well-written and easy to follow and understand. The Introduction includes all the data needed for readers to have an image of the field. The methods are well presented. The results follow the main aspects: demographic data, margin status, survival analysis, prognostic factors, complications, and adjuvant therapy. Survival outcomes were analyzed based on the margin status, type of surgery, complications, and adjuvant therapy. Their results are essential for the practical side.
I would recommend to change the title of the manuscript. It seems too long and challenging.
Also, the authors may include a small paragraph on the strengths of their research in the discussion.
I would present the graphs with the survival outcomes larger to be easily understood, a maximum of 2 on a line but larger than they are in the manuscript.
Author Response
Reviewer1.
The authors present their experience with the different types of surgery for bile duct cancer, considering the margin status, complications, and adjuvant therapy.
The manuscript is well-written and easy to follow and understand. The Introduction includes all the data needed for readers to have an image of the field. The methods are well presented. The results follow the main aspects: demographic data, margin status, survival analysis, prognostic factors, complications, and adjuvant therapy. Survival outcomes were analyzed based on the margin status, type of surgery, complications, and adjuvant therapy. Their results are essential for the practical side.
#I would recommend to change the title of the manuscript. It seems too long and challenging.
☞Author’s reply
: Thank you for this critical comment. We change the title into “Survival comparison of different operation type for middle bile duct cancer considering surgical complication and adjuvant treatment effects”.
#Also, the authors may include a small paragraph on the strengths of their research in the discussion.
☞Author’s reply
: Thank you for pointing this part. We add a paragraph addressing strengths of our research at the end of the discussion part (line 313-317)
#I would present the graphs with the survival outcomes larger to be easily understood, a maximum of 2 on a line but larger than they are in the manuscript.
☞Author’s reply
: Thank you for this advice. We edited our graph larger to improve readability.
**The revised manuscript file is sent in two versions. One is an edited file without any special mark, and the other one is an edited file with red mark. The edited part was changed to red. Thank you for your consideration. I look forward to hearing from you.
Reviewer 2 Report
Comments and Suggestions for Authors
thank you for allowing me to review this retrospective monocentric study which included more than 500 patients with cholangiocarcinoma of the main bile duct over 10 years. this is a substantial series of patients. the authors compared the oncological results of two surgical procedures; the strategy being unclear in this location.
in the introduction chapter :
what is the incidence of cholangiocarcinoma of the trunk of the main bile duct; the term "rare" is not clear.
the authors report the scarcity of studies on this subject, without citing any references: how many studies? how many patients included?
as far as the aim of the study is concerned, it's more a question of comparison than of determining the most appropriate surgical strategy. if not, the study design should be changed to a randomized trial or propensity score.
in the materials and methods chapter
can you provide the reference of the ethics committee?
the authors used the Dindo-Clavien classification for the severity of post-operative complications. what time frame was used?
how did you design your multivariate analysis? what was the percentage of missing data?
in the results section
the management of this type of cancer may, in a number of cases, require pre-operative drainage of the bile duct, which may influence the time required for surgery and morbidity. however, this variable was not taken into account. why not?
in univariate analysis, R0/R1 had a small p<0.20 (2.108 0.783-5.678 0.131). Why was this data not included in a multivariate statistical model, given its prognostic importance in the literature?
what is the post-operative mortality rate in this series?
what was the strategy in the event of a positive extemporaneous margin? how many patients initially scheduled for one type of surgery had additional surgery such as duodenopancreatectomy and/or hepatectomy? were the results then analyzed on an intention-to-treat basis?
in the discussion section
the paragraph between lines 259 and 264 refers to the results, yet it appears in the discussion section without any information in the results section?
according to the literature, resection of the bile duct would not be the right strategy in cases of lymph node invasion of relays 13, 14 and 17. should lymph node picking with extemporaneous examination be performed before the operation?
Author Response
Reviwer2.
thank you for allowing me to review this retrospective monocentric study which included more than 500 patients with cholangiocarcinoma of the main bile duct over 10 years. this is a substantial series of patients. the authors compared the oncological results of two surgical procedures; the strategy being unclear in this location.
in the introduction chapter
#what is the incidence of cholangiocarcinoma of the trunk of the main bile duct; the term "rare" is not clear.
☞Author’s reply
: Thank you for pointing unclear words. Saha et al. reported that the incidence of intrahepatic cholangiocarcinoma is reported from 0.44 to 1.18 cases per 100,000, while the incidence of extrahepatic cholangiocarcinoma is from 0.95 to 1.02 per 100,000, during the 40-year period. We will add this sentence and reference in the introduction part. (line 49-51)
Saha SK, Zhu AX, Fuchs CS, Brooks GA. Forty-Year Trends in Cholangiocarcinoma Incidence in the U.S.: Intrahepatic Disease on the Rise. Oncologist. 2016 May;21(5):594-9. doi: 10.1634/theoncologist.2015-0446. Epub 2016 Mar 21. PMID: 27000463; PMCID: PMC4861366.
#the authors report the scarcity of studies on this subject, without citing any references: how many studies? how many patients included?
☞Author’s reply
: Thank you for the comment. Four studies were conducted on the same topic, each involving data from 194, 96, 184, 105 patients, respectively. The references were written below. However, to the best of our knowledge, this study was the first one compare between two operations considering comprehensively with surgical margin, adjuvant treatment, and complication at the same time.
Lee HG, Lee SH, Yoo DD, Paik KY, Heo JS, Choi SH, Choi DW. Carcinoma of the middle bile duct: is bile duct segmental resection appropriate? World J Gastroenterol. 2009 Dec 21;15(47):5966-71. doi: 10.3748/wjg.15.5966. PMID: 20014461; PMCID: PMC2795184.
Kim N, Lee H, Min SK, Lee HK. Bile duct segmental resection versus pancreatoduodenectomy for middle and distal common bile duct cancer. Ann Surg Treat Res. 2018 May;94(5):240-246. doi: 10.4174/astr.2018.94.5.240. Epub 2018 Apr 30. PMID: 29732355; PMCID: PMC5931934.
Schreuder AM, Engelsman AF, van Roessel S, Verheij J, Besselink MG, van Gulik TM, Busch OR. Treatment of mid-bile duct carcinoma: Local resection or pancreatoduodenectomy? Eur J Surg Oncol. 2019 Nov;45(11):2180-2187. doi: 10.1016/j.ejso.2019.06.032. Epub 2019 Jun 29. PMID: 31279596.
Akita M, Ajiki T, Ueno K, Tsugawa D, Tanaka M, Kido M, Toyama H, Fukumoto T. Benefits and limitations of middle bile duct segmental resection for extrahepatic cholangiocarcinoma. Hepatobiliary Pancreat Dis Int. 2020 Apr;19(2):147-152. doi: 10.1016/j.hbpd.2020.01.002. Epub 2020 Jan 30. PMID: 32037277.
#as far as the aim of the study is concerned, it's more a question of comparison than of determining the most appropriate surgical strategy. if not, the study design should be changed to a randomized trial or propensity score.
☞Author’s reply
: This is a critical point that we need to address further. The primary objective of this study was to identify an optimized treatment method for individual patients. While conducting the research as a randomized controlled trial could yield results aligned with this goal, ethical concerns may arise. Additionally, due to the limited size of patient groups, deriving meaningful results through propensity matching becomes challenging. As a following to this study, we intend to pursue a multicenter investigation on the same subject, and thanks to your advice, we will design our research as propensity matching.
in the materials and methods chapter
#can you provide the reference of the ethics committee?
☞Author’s reply
: Thank you for this comment. This study was retrospectively conducted through data analysis, adhering to the review process of the Institutional Review Board (IRB) at Samsung Medical Center. The corresponding IRB approval documentation will be provided as an attachment.
#the authors used the Dindo-Clavien classification for the severity of post-operative complications. what time frame was used?
☞Author’s reply
: Thank you for this comment. Patients were categorized according to the Clavien-Dindo classification, based on complications occurring within 90 days following surgery. The postoperative complication time frame is added on the manuscript. (line 127-128)
#how did you design your multivariate analysis? what was the percentage of missing data?
☞Author’s reply
: Thank you for this comment. Multivariate analysis of independent prognostic factors for OS was identified by using the Cox proportional hazard model. All tests were two-sided and a p-value under 0.1 was considered statistically significant. There were no missing data in this study. The design of uni-and multivariate analysis was added in the manuscript (2.5 Statistical Analysis, line 149-152)
in the results section
#the management of this type of cancer may, in a number of cases, require pre-operative drainage of the bile duct, which may influence the time required for surgery and morbidity. however, this variable was not taken into account. why not?
☞Author’s reply
: Thank you for this critical comment. We conducted a univariate analysis involving preoperative biliary drainage, revealing a hazard ratio of 1.737 (95% confidence interval: 1.222-2.467, p=0.002). Despite the statistical significance observed in the univariate analysis, preoperative biliary drainage was excluded from the multivariate analysis due to its reflection of tumor size. In the group whom inevitably underwent biliary drainage before surgery, their biliary obstruction sign occurred as a result of the cancer's mass effect, which creating overlapping effect with the T-stage in the multivariate analysis.
#in univariate analysis, R0/R1 had a small p<0.20 (2.108 0.783-5.678 0.131). Why was this data not included in a multivariate statistical model, given its prognostic importance in the literature?
☞Author’s reply
: This is a critical point that we need to address further. Despite emphasizing the significance of the resection margin in our argument, its omission from the multivariate analysis can be attributed to two primary factors. First, there were only six margin positive patients, which were very limited number of cases. Second, since the influence of N-stage’s statistical power has robust, magnitude of margin was shown relatively minor impact In this study, a p-value of 0.1 was deemed as the threshold for determining statistical significance in the multivariate analysis. As suggested by the reviewer, if the p-value threshold were set at 0.2, the results of the multivariate analysis would be as follows. However, even with the elevated threshold, still the resection margin was not statistically significant in the multivariate analysis, which is a critical limitation of this study. Because of this reason, we didn’t change the Table 2.
Table 2. Prognostic Factors for Overall Survival in Patients with Middle Bile Duct Cancer.
|
Variable |
|
Patients (n) |
5Y OS (%) |
Univariate analysis |
Multivariate analysis |
||||
|
HR |
95% CI |
p-value |
HR |
95% CI |
p-value |
||||
|
Sex |
male/female |
342/178 |
52.5/57.2 |
0.871 |
0.676-1.122 |
0.285 |
|
|
|
|
Age |
≤ 65 / > 65 |
229/291 |
65.4/44.7 |
2.087 |
1.619-2.690 |
<0.001 |
2.164 |
1.660-2.821 |
<0.001 |
|
BMI |
≤ 25 / > 25 |
378/142 |
50.9/63.2 |
0.769 |
0.579-1.021 |
0.070 |
0.831 |
0.624-1.106 |
0.204 |
|
ASA score |
I |
82 |
57.1 |
|
|
0.176 |
|
|
0.966 |
|
II |
395 |
54.6 |
1.133 |
0.817-1.572 |
0.455 |
0.999 |
0.718-1.391 |
0.831 |
|
|
III/IV |
43 |
43.7 |
1.567 |
0.969-2.534 |
0.067 |
1.055 |
0.644-1.728 |
0.027 |
|
|
Preop CA19-9 |
≤ 35 / > 35 |
265/255 |
62.3/45.9 |
1.539 |
1.210-1.958 |
<0.001 |
1.321 |
1.033-1.691 |
0.027 |
|
T-stage |
T1&2/T3&4 |
448/72 |
58.4/28.0 |
2.361 |
1.761-3.164 |
<0.001 |
1.658 |
1.219-2.255 |
0.001 |
|
N-stage |
N(-)/N(+) |
359/161 |
62.4/35.5 |
2.154 |
1.689-2.747 |
<0.001 |
2.019 |
1.558-2.616 |
<0.001 |
|
Operation |
BDR / PD |
131/389 |
54.7/54.0 |
0.929 |
0.707-1.219 |
0.594 |
|
|
|
|
Resection margin |
R0 / R1 |
514/6 |
54.4/33.3 |
2.108 |
0.783-5.678 |
0.131 |
1.330 |
0.701-2.523 |
0.383 |
|
Complicationsa |
no / yes |
389/131 |
55.3/50.3 |
1.046 |
0.791-1.383 |
0.752 |
|
|
|
|
Adjuvant treatment |
no / yes |
403/117 |
55.6/49.1 |
1.077 |
0.806-1.437 |
0.617 |
|
|
|
aMajor complication indicated Clavien-Dindo grade ≥ 3.
OS, overall survival; PD/PPPD, pancreaticoduodenectomy/pylorus-preserving pancreaticoduodenectomy; BDR, Bile duct resection.
#what is the post-operative mortality rate in this series?
☞Author’s reply
: Thank you for this comment. In this study, the postoperative mortality rate was 0.8%.
#what was the strategy in the event of a positive extemporaneous margin? how many patients initially scheduled for one type of surgery had additional surgery such as duodenopancreatectomy and/or hepatectomy? were the results then analyzed on an intention-to-treat basis?
☞Author’s reply
: Thank you for this critical comment. In all cases of mid-bile duct cancer, the initial intention is bile duct resection, but bile duct resection is always attempted with the possibility that the surgery may become extended. In the case of a positive margin on frozen biopsy during bile duct resection, a decision to perform an extended operation is made in consideration of the patient's comorbidity. Due to the suspicion of positive margin on frozen biopsy, 12.7% of patients underwent additional pancreas resection, and 20.6% of patients underwent additional liver resection. In the case of this retrospective study, the classification was not based on initial intention, but based on the final surgical method.
in the discussion section
#the paragraph between lines 259 and 264 refers to the results, yet it appears in the discussion section without any information in the results section?
☞Author’s reply
: Thank you for this comment. We will add the description of supplemental table1 in the result section (3.1 Demographics and margin status, line 167-169)
#according to the literature, resection of the bile duct would not be the right strategy in cases of lymph node invasion of relays 13, 14 and 17. should lymph node picking with extemporaneous examination be performed before the operation?
☞Author’s reply
: Thank you for this critical comment. In the context of bile duct cancer, the efficacy of anticancer drugs is less established in comparison to other cancer types. Given that surgical resection remains the foremost validated treatment, rather than extemporaneous examination, pancreaticoduodenectomy is performed when there is suspicion of positive lymph nodes in stations 13, 14, and 17. This suspicion can be confirmed through excision.
**The revised manuscript file is sent in two versions. One is an edited file without any special mark, and the other one is an edited file with red mark. The edited part was changed to red. Thank you for your consideration. I look forward to hearing from you.
Round 2
Reviewer 1 Report
Comments and Suggestions for Authors
The authors made the changes suggested by the reviewer. I have mo other comments regarding the manuscript.
Reviewer 2 Report
Comments and Suggestions for Authors
the authors have responded point by point to questions and comments that have significantly improved the quality of the manuscript